# Peripherally Restricted Activation of Opioid Receptors Influences Anxiety-Related Behaviour and Alters Brain Gene Expression in a Sex-Specific Manner

**DOI:** 10.3390/ijms252313183

**Published:** 2024-12-07

**Authors:** Nabil Parkar, Wayne Young, Trent Olson, Charlotte Hurst, Patrick Janssen, Nick J. Spencer, Warren C. McNabb, Julie E. Dalziel

**Affiliations:** 1AgResearch, Palmerston North 4410, New Zealand; nabil.parkar@queensu.ca (N.P.); wayne.young@fonterra.com (W.Y.); trent.olson@agresearch.co.nz (T.O.);; 2Riddet Institute, Massey University, Palmerston North 4410, New Zealand; w.mcnabb@massey.ac.nz; 3School of Food and Advanced Technology, Massey University, Palmerston North 4410, New Zealand; 4College of Medicine and Public Health, Flinders Health & Medical Research Institute, Bedford Park, Adelaide, SA 5042, Australia; nicholas.spencer@flinders.edu.au

**Keywords:** enteric nervous system, loperamide, colon, hippocampus, amygdala, microbiota, gut–brain axis, depression

## Abstract

Although effects of stress-induced anxiety on the gastrointestinal tract and enteric nervous system (ENS) are well studied, how ENS dysfunction impacts behaviour is not well understood. We investigated whether ENS modulation alters anxiety-related behaviour in rats. We used loperamide, a potent μ-opioid receptor agonist that does not cross the blood–brain barrier, to manipulate ENS function and assess changes in behaviour, gut and brain gene expression, and microbiota profile. Sprague Dawley (male/female) rats were acutely dosed with loperamide (subcutaneous) or control solution, and their behavioural phenotype was examined using open field and elevated plus maze tests. Gene expression in the proximal colon, prefrontal cortex, hippocampus, and amygdala was assessed by RNA-seq and caecal microbiota composition determined by shotgun metagenome sequencing. In female rats, loperamide treatment decreased distance moved and frequency of supported rearing, indicating decreased exploratory behaviour and increased anxiety, which was associated with altered hippocampal gene expression. Loperamide altered proximal colon gene expression and microbiome composition in both male and female rats. Our results demonstrate the importance of the ENS for communication between gut and brain for normo-anxious states in female rats and implicate corticotropin-releasing hormone and gamma-aminobutyric acid gene signalling pathways in the hippocampus. This study also sheds light on sexually dimorphic communication between the gut and the brain. Microbiome and colonic gene expression changes likely reflect localised effects of loperamide related to gut dysmotility. These results suggest possible ENS pharmacological targets to alter gut to brain signalling for modulating mood.

## 1. Introduction

The complex bidirectional communication system of the gut–brain axis (GBA) not only ensures the body responds appropriately to stress but also permits signalling from the gut to influence mood, higher cognitive functions, and mental health [1,2]. Peripheral neural pathways from the enteric nervous system (ENS), involving both spinal and vagal afferents, form critical components of the gut–brain axis [3]. The ENS, deeply embedded within smooth muscle layers of the gut wall, consists of a diverse population of enteric neurons and glial cells that orchestrate GI functions including motility and secretion [4,5]. The ENS also communicates with the brain via chemical signalling pathways, allowing gut luminal factors to indirectly influence brain function.

The gut microbiome impacts development and function of both the ENS and CNS [6,7]. For example, in germ-free (GF) mice (specially raised animals devoid of all microorganisms) the number of enteric neurons is reduced, alongside abnormal GI motility [8,9]. In addition, changes in microbiota composition or dysbiosis have been shown to influence cognitive function, mood, behaviour, and brain neurochemistry [10,11,12,13]. Given the proximity of the gut microbiome to the ENS, the gut microbiome has surprisingly not emerged as a major influence on the ENS, with subsequent effects on GI physiology and potentially gut to brain signalling [14,15].

Opioid receptor subtypes mu, delta, and kappa are widely distributed in the brain and GI tract neural pathways [16,17]. These receptors can be activated endogenously by endorphins, dynorphins, enkephalins, and endomorphins as well as exogenously by opioid agonists. Brain opioid receptors regulate pain, emotion, stress, feeding motivation, and eating behaviour [18]. In the gut, endogenous opioid peptides, including enkephalins, dynorphins, and endorphins, regulate gastrointestinal function by slowing motility and secretomotor activity [19]. Mu opioid receptors expressed in the gut can regulate the gut to brain neural circuitry involved in satiety [20,21]. For example, ingested soymorphin (soy-derived mu opioid peptide) suppresses food intake via activation of gut mu opioid receptors [21].

The aim of this study was to investigate whether manipulating ENS activity affects anxiety-related behavioural responses to reveal key insights into the mechanisms of gut to brain signalling. We used a peripherally acting opioid agonist, loperamide, due to its inability to cross the blood–brain barrier [22,23]. This enabled us to investigate how peripheral neural stimulation might influence brain neurochemical pathways and anxiety-related behaviour. Loperamide (Imodium™) is an antimotility drug that activates ENS opioid receptors primarily of the mu type, and also delta and kappa but with lower affinity [24], in the myenteric plexus of the large intestine, decreasing activity of the myenteric plexus, leading to a reduction in tone of the longitudinal and circular smooth muscles and a slowing of GI transit [25,26]. We assessed the effect of acute dosing with loperamide on rat behaviour in open field and elevated plus maze tests and measured changes in gene expression within specific brain regions that play critical roles in stress responses, namely the amygdala, hippocampus, and prefrontal cortex. To further elucidate the possible influence of ENS mu opioid receptor activation on gut to brain signalling, changes in proximal colon gene expression and caecal microbiome composition were also determined.

## 2. Results

### 2.1. Animal Metrics

The study design and experimental schedule followed for measurements and sampling are outlined in Figure 1 (refer to Section 4.3). Loperamide reduced food intake by 38% in male and 44% in female rats, 24 h posttreatment, compared to pretreatment controls (Table 1). Loperamide also reduced faecal output by 31% in male and 38% in female rats, compared to pretreatment controls. No change in bodyweight occurred in any of the treatment groups. Statistical analysis using two-way repeated measures ANOVA showed main effects of loperamide treatment (F(1,27) = 33.29, *p* < 0.0001) and sex (F(1,27) = 47.39, *p* < 0.0001) for food intake, and loperamide treatment (F(1,27) = 21.22, *p* < 0.0001) and sex (F(1,27) = 44.00, *p* < 0.0001) for faecal output. There was no interaction between sex and treatment for any of the animal metrics.

### 2.2. Behaviour Tests

#### 2.2.1. Open Field Test

Total distance moved was reduced by 30% in loperamide-treated female rats compared to controls (Figure 2a). Similarly, loperamide-treated female rats moved 30% more slowly than controls (Figure 2b). Loperamide treatment resulted in less rearing against the walls in female rats (Figure 2c). In contrast, loperamide did not significantly alter total distance moved, velocity, or rearing in males. Loperamide did not alter the time spent in different zones of the open field arena (Figure 2d) for either male or female rats (Figure 2e,f). Statistical analysis using two-way repeated measures ANOVA showed a main effect of loperamide treatment for distance moved (F(1,27) = 10.29, *p* = 0.0034) and for velocity (F(1,27) = 10.29, *p* = 0.0034). Main effects of treatment (F(1,27) = 10.73, *p* = 0.0029) and sex (F(1,27) = 24.86, *p* < 0.0001) were found for rearing frequency.

#### 2.2.2. Elevated Plus Maze

Loperamide-treated female rats moved 21% less (Figure 3a) and velocity was 21% slower compared to controls (Figure 3b). No differences were detected in entries to open arms between loperamide-treated female rats and controls (*n* = 15) (Figure 3c). However, the time spent in the open arms of the EPM (Figure 3d) was decreased by 24% in loperamide-treated females compared to controls (Figure 3e). Time spent in and entries to closed arms did not differ between loperamide-treated females and controls. Loperamide did not affect performance of male rats for any of the EPM parameters compared to controls. No significant differences were observed in time spent in the center zone of the EPM between loperamide-treated male and female rats and their control counterparts (Figure 3f). Statistical analysis using two-way repeated measures ANOVA showed main effects of loperamide treatment (F(1,27) = 8.804, *p* = 0.0062) and sex (F(1,27) = 4.900, *p* = 0.0355) for distance moved in the EPM, in addition to an interaction between loperamide treatment and sex (F(1,27) = 5.686, *p* = 0.0244). Main effects of loperamide treatment (F(1,27) = 8.811, *p* = 0.0062) and sex (F(1,27) = 4.890, *p* = 0.0355) were found for velocity in the EPM, in addition to an interaction between loperamide treatment and sex (F(1,27) = 5.686, *p* = 0.0244). A main effect of loperamide treatment (F(1,27) = 4.536, *p* = 0.0425) but not sex (F(1,27) = 0.1262, *p* = 0.7252) was found for time spent in the open arms, in addition to an interaction between loperamide treatment and sex (F(1,27) = 4.265, *p* = 0.0486).

### 2.3. Brain Gene Expression

In female rats, nine genes in the hippocampus were found to be decreased in expression between control and loperamide treatment groups (Figure 4a, Table 2, Appendix A), but there were no differences at the individual transcript level in the amygdala or prefrontal cortex. In male rats, there were no differences in gene expression between treatment groups at the individual transcript level in any of the brain regions.

While the analysis of individual genes showed significant differences only in expression levels in the hippocampus of female rats, gene set enrichment analysis (GSEA) showed that the collective expression patterns of genes within numerous reactome pathways also differed in the prefrontal cortex and amygdala between groups (Figure 4b). The greatest number of differentially expressed pathways were seen in the hippocampus (33 pathways) which included several immune-signalling-related pathways involving TAK1, NOD1/2, and CD28, showing higher expression in loperamide-treated rats, while pathways related to GABA synthesis, release, reuptake, and degradation showed a significant overall decrease in expression in females. In the prefrontal cortex, seven pathways were differentially expressed, with most showing higher expression in the loperamide-treated female group. Two pathways showed lower expression in loperamide-treated female rats compared to controls, MAP kinase activation and interleukin 17 signalling. In the amygdala, pathways related to ERK and NOD1/2 showed increased expression in loperamide-treated female rats.

### 2.4. Proximal Colon Gene Expression

Overall, more than 200 genes of the proximal colon were differentially expressed between control and loperamide groups (Appendix A). Ninety-five genes were more highly expressed in the loperamide group compared to controls (Figure 5a). The differentially expressed genes between control and loperamide treatment groups in both sexes were mainly associated with biological processes such as response to corticosterone, cellular response to glucocorticoid stimulus, and various signal-related pathways such as the lipopolysaccharide-mediated signalling pathway, phospholipase C-activating G protein-coupled receptor signalling pathway, and peroxisome proliferator activated receptor (PPAR) signalling pathway. In terms of sex differences, 145 genes were differentially expressed between control and loperamide groups in male rats, whereas 60 genes were differentially expressed between control and loperamide groups in female rats. The top 40 differentially expressed genes between control and loperamide groups in the proximal colon are shown for female and male rats (Figure 5b,c, Appendix A).

### 2.5. Caecal Microbiome

#### 2.5.1. Microbiota at the Family and Genus Level

Decreased faecal output by loperamide in both male and female rats indicated that GI transit was prolonged. We therefore evaluated whether caecal microbiota composition was also affected. Microbiota profiles differed significantly between control and loperamide-treated groups in both male and female rats (Table 3). However, there were no significant differences between male and female rats, nor was there a significant interaction between sex and treatment.

Analysis of the microbiotas at the family level showed significant differences (*q* < 0.05) between the control and loperamide-treated groups in ten families (Figure 6a). Of these, the majority were from the Firmicutes phylum: *Lactobacillaceae*, *Clostridiaceae*, *Eubacteriaceae*, *Lachnospiraceae*, *Ruminococcaceae.* Except for *Lactobacillaceae*, all other families from the Firmicutes phylum were less abundant in the loperamide-treated group. Four families from the Bacteroidetes phylum, *Bacteroidaceae*, *Barneseillaceae*, *Muribaculoceae*, and *Tannerellaceae*, were more abundant in the loperamide-treated group.

Principal coordinate analysis of UniFrac phylogenetic distances showed strong separation overall between caecal microbiotas from control and loperamide-treated male and female rats (Figure 6b).

At the genus level, 20 genera differed significantly between treatments. Genera that were significantly more abundant in the loperamide-treated group and that had a mean relative abundance greater than 1% included *Bacteroides*, *Lactobacillus*, *Butyricicoccus*, *Clostridium*, *Colidextribacter*, *Eubacterium*, *Acetatifactor*, *Blautia*, *Dorea*, *Enterocloster*, *Roseburia*, *Schaedlerella*, *Flavonifractor*, and *Ruminococcus*. The largest effect of treatment on the microbiota composition was seen in *Bacteroides* and *Lactobacillus*. Loperamide treatment increased the relative abundance of *Bacteroides* by 75% in male rats and 32% in female rats compared to controls (Figure 6c, Table 3). Similarly, the abundance of *Lactobacillus* increased in loperamide-treated male and female rats by 171% and 100%, respectively, compared to controls.

#### 2.5.2. Metagenome Community Function

In addition to characterising changes in the caecal microbiota composition, the functional characterisations of the caecal microbiome among different treatment groups were compared. Functional annotations were carried out for the updated gene catalogue using the KEGG database. In male rats, at the third classification level, the top five KEGG pathways based on significance (*q* < 0.05) that showed significant changes in abundance when comparing the control and loperamide groups were K1000240 Pyrimidine metabolism, K1000230 Purine metabolism, K1000010 Glycolysis/Gluconeogenesis, K1002020 Two-component system, and K1000300 Lysine biosynthesis (Table 4). These reference pathways were associated with nucleotide metabolism, carbohydrate metabolism, signal transduction, and amino acid metabolism. In female rats, the top five KEGG pathways based on significance (*q* < 0.05) that showed significant changes in abundance when comparing the control and loperamide groups included K1000361 Chlorocyclohexane and chlorobenzene degradation, K1000230 Purine metabolism, K1000040 Pentose and glucuronate interconversions, K1000040 Pentose and glucuronate interconversions, K1000480 Glutathione metabolism, and K1000052 Galactose metabolism. These reference pathways were associated with xenobiotics biodegradation and metabolism, nucleotide metabolism, carbohydrate metabolism, and metabolism of other amino acids.

### 2.6. Dataset Integration

Dataset integration showed that duration in the EPM closed arms correlated with being in the outer zone of the OFT (higher anxiety) and with expression of three hippocampal mitochondrial genes involved in energy metabolism: cytochrome b (Mt-cyb), ATP synthase (Mt-atp6), and cytochrome c oxidase subunit I (Mt-co1), a component of cytochrome c oxidase (Figure 7). EPM closed arm duration also correlated with proteolipid protein 1 (Plp1; PFC), a gene encoding a transmembrane proteolipid protein that is the predominant myelin protein important in brain development, and hippocalcin (Hpca; AMY), a calcium-binding protein in a neuronal calcium sensor. In the proximal colon, EPM closed arm duration correlated with expression of three genes: serine protease inhibitor Kazal type 1 (Spink4), an inflammatory growth factor and tumorigenic marker, the secreted gel-forming mucin 2 (Muc2), which forms a mucus layer barrier on the colon wall, and regenerating family member 4 (Reg4), involved in infection and inflammation. EPM closed arm duration positively correlated with Bifidobacterium, Barnesiella, and unclassified Odoribacteraceae and negatively correlated with unclassified Paenibacillaceae and the KEGG K22704 hpnR (hopanoid C-3 methylase) pathway. Likewise, Bifidobacterium, Barnesiella, and unclassified Odoribacteraceae were positively correlated with the hippocampal mitochondrial genes.

## 3. Discussion

The main finding from this study that peripheral opioid receptor activation altered anxiety-related behaviour indicates that opioid receptors play a role in gut to brain signalling. Loperamide primarily activates mu opioid agonists in the ENS but will also have acted on delta and kappa receptors located on enteric neurons throughout the rat GI tract [26], and these will likely also have contributed to the opioid-receptor-mediated effects. The behavioural changes induced by loperamide were female specific and associated with changes in hippocampal gene expression. In contrast, the loperamide-induced shifts in the caecal microbiome were not sex specific and occurred in a relatively short timeframe, just 48 h after acute dosing, showing rapid shifts in community composition are possible following slowed colonic transit. Proximal colon gene expression had some sex specificity but was not strongly correlated with brain function changes.

Reduced food intake by loperamide treatment together with decreased faecal output indicated that this dose was effective at altering ENS activity. Activation of peripheral, predominantly gastric, opioid receptors leads to suppression of food intake in rats and was expected [18]. The fact that we observed reduced food intake is consistent with a peripheral origin of action for loperamide, because activation of brain mu opioid receptors instead increases eating behaviour in rodents [27].

### 3.1. Behavioural Changes with Loperamide

The lower percentage of time spent in open arms in loperamide-treated female rats (than controls) indicated heightened anxiety. The suppression of locomotor activity by loperamide in female rats and not male rats (OF and EPM) can be attributed to increased anxiety. The lack of loperamide’s effect in males contrasts with a report in which, 30 min following a single dose of intragastrically administered loperamide (5 mg/kg) in male rats, time spent and entries in the open arms of the EPM increased, suggesting an anxiolytic effect of loperamide [28]. However, this was not reported relative to the closed arms according to convention. In any case, the apparent discrepancy with our findings may stem from elevated baseline stress as corticosterone can remain high for 4 h postgavage [29], as opposed to more than 60% recovered 1.5 h following S.C. injection [30]. We are not aware of prior studies that have investigated the effects of loperamide on rodents in the open field test, however, effects of other opioids have been investigated [31,32]. Morphine belongs to the same class of drug, but unlike loperamide [23], morphine crosses the blood–brain barrier, acting centrally via dopaminergic systems [33,34]. Nevertheless, peripheral administration of morphine has been shown to induce dose-dependent biphasic effects on locomotor activity [32]. A study in male rats showed that a low dose of morphine reduced distance moved in the open field test, indicating suppressed locomotor activity, but a higher dose induced hyperlocomotion in the OF [32].

We cannot rule out the possibility that the increase in anxiety-like behaviour is the result of decreased locomotor activity. However, it seems unlikely that loperamide at this dose would have acted centrally to cause sedation and have affected only females. While morphine can induce a locomotor suppressor effect, this occurs for both sexes and the effect is greater in males than in females [35].

### 3.2. Gene Expression in Relation to Behaviour

Rearing behaviour in the OF test gave an additional measure of anxiety providing further information indicative of exploratory behaviour associated with information gathering and cognition [36]. It consists of rats standing on both hind paws in a vertical upright position and can be classified as unsupported and supported rears [37]. The neurotransmission of gamma-aminobutyric acid (GABA) controlled by the GABA_A_ receptor in the hippocampus is closely linked to rearing behaviour [38]. Notably, the gene set enrichment analysis unveiled a significant downregulation of the GABA synthesis, release, reuptake, and degradation pathway in the hippocampus of loperamide-treated female rats compared to the control group. The hippocampus responds to stress as a conflict resolution and inhibitory control center [39]. Our finding that females displayed a higher frequency of supported rearing compared to males at baseline are in line with a study by Zhan and colleagues where female mice (controls) exhibited higher rearing frequency compared to their male counterparts [40]. This suggests females have higher levels of exploratory tendencies compared to males. Sex differences may relate to ovarian hormones affecting hippocampal function and morphology [41] and impact hippocampus-dependent behaviors [42,43]. In contrast to the increased rearing frequency in control females, our findings revealed that loperamide significantly reduced rearing frequency in females, but no differences were observed in males. Although there are no studies on the effect of loperamide on rearing behaviour in rodents, our results are in accordance with a study reporting that morphine decreases rearing activity in female mice [31].

The simplest explanation for the increased anxiety induced by loperamide in females is that the activation of peripheral mu opioid receptors inhibited ENS firing and vagal afferent firing and reduced activation of brain pathways associated with GABA inhibition (and possibly the neuroimmune axis), promoting neural excitation and leading to increased anxiety. The ability of loperamide to disrupt propagating contractions in the proximal and midcolon regions indicates that what happens in the proximal colon is very important and may have a bearing on the results presented here [44]. The 60 genes altered in proximal colon expression in females suggest that gut barrier/immunomodulatory pathways are affected.

The fact that brain gene expression was only altered in female rats supports these changes as contributing to the behavioural differences observed. Because loperamide does not cross the blood–brain barrier, we infer that this is a peripherally induced effect via the gut–brain axis. Our results demonstrate that loperamide administration led to a reduction in corticotropin releasing hormone receptor 2 (CRHR2) gene expression in the hippocampus of female rats compared to controls. CRHR2 is one of the receptors through which corticotropin-releasing hormone (CRH) acts to regulate stress responses and plays an important role in the development of anxiety and depression [45,46]. For example, CRHR2 knockout male and female mice exhibit increased anxiety in the EPM where mice spent less time in and entered the open arms less frequently [47]. CRHR2 appears to play a role in reducing anxiety-like behaviour, and the downregulation of CRHR2 in loperamide-treated female rats may contribute to the anxiogenic effects seen in the behavioural tests. Furthermore, our study also points to a possible role of the peripheral opioid system in modulating the expression of CRHR2, implicating this in GBA regulation.

### 3.3. Microbiota Composition and Function

The rapid shift in caecal microbiota profile in response to peripheral opioid receptor activation in both sexes is a likely response to slowed colonic transit. Although it is unlikely that a microbiota factor alone is responsible for the behavioural effect observed in females, the contributory effects of microbiota on behavioural outcomes should not be ruled out. The prolonged transit time induced by loperamide could have contributed to the corresponding rise in the relative abundance of Bacteroidetes as members of this group, including *Bacteroides*, are specialist fibre degraders that would be advantaged by having more time to break down difficult to degrade carbohydrates [48,49]. Moreover, certain gut microbes possess the ability to adapt their substrate utilisation efficiency according to the availability of resources [50,51]. For example, *Bacteroides* can adjust their metabolism and metabolic products through a metabolic switch [51]. This adaptability grants them a competitive advantage, enabling them to thrive and outcompete other bacteria, particularly during periods of reduced substrate availability. On the other hand, the decelerated transit caused by loperamide resulted in a decrease in the relative abundance of the genus *Roseburia*, which is typically associated with faster colonic transit [52]. These outcomes indicate that *Roseburia* may not thrive well in a slowed transit luminal environment. Collectively, these findings suggest that maintaining normal gut motility is crucial for preserving a balanced gut ecosystem and gut homeostasis.

Functional characterisations of the caecal microbiome using the KEGG database revealed sex-specific effects of loperamide on microbiome function and show differential responses and protective mechanisms in response to this treatment. In females, loperamide increased the abundance of genes related to xenobiotic and glutathione metabolism in pathways crucial for detoxification of harmful substances that provides a defence mechanism for the microbes [53], perhaps a microbial response to the change in environmental conditions resulting from slowed transit. In males, loperamide decreased the relative abundance of genes involved in the two-component system, integral to the bacterial quorum sensing mechanism [54]. This suggests that in males loperamide may have a disruptive effect on communication and interaction between microbial communities, while no significant impact occurred in females.

The positive correlation between anxiety and Bifidobacterium is expected as it has a role in regulation of anxiety, mood, and cognition in rodents and humans, and Barnesiella has been positively associated with anxiety and depression [55] and Odoribacteraceae with depression [56]. The positive correlation among anxiety, hippocampal energy metabolism genes, and bacterial taxa revealed by integrated dataset analysis implicates gut to brain signalling in the increased anxiety driven by loperamide treatment. A lower brain mitochondria respiration rate has been reported in highly anxious rats [57,58] and is consistent with our findings that time spent in the closed arms positively correlated with expression of three hippocampal mitochondrial genes (Mt-cyb, Mt-atp6, Mt-co1) involved in energy metabolism. Human studies show increasing evidence for a link between mitochondrial function and stress-related anxiety [59,60]. While the association between anxiety and colonic wall genes involved in barrier protection also suggests gut to brain signalling, the lack of direct association between colon genes and bacterial taxa suggests their relationships with anxiety were largely independent. The correlation between anxiety and decreased hippocalcin myelin protein expression in the amygdala might reflect the role of hippocalcin in neuronal excitability as a diffusible calcium sensor [61].

## 4. Materials and Methods

### 4.1. Ethical Approval

This study was approved (AE15077) by the AgResearch Grasslands Animal Ethics Committee (Palmerston North, New Zealand) and carried out in accordance with the Animal Welfare Act, 1999 (NZ).

### 4.2. Animals

The Sprague Dawley rat strain was selected because it is considered normo-sensitive in its stress response and therefore a suitable human model. Both male and female rats were used to assess possible sex differences in stress response to behavioural tests. Adult Sprague Dawley rats, 16 male and 16 female, were 6–8 weeks of age (200–300 g) (AgResearch Small Animal Breeding Unit, Ruakura, Hamilton, New Zealand). The animals were pair-housed (as required by animal ethics) in sawdust-lined Techniplast plastic/stainless steel cages. Rats were acclimatised for a week and pair-housed at 22 °C and 55% humidity and maintained under a 12 h light/dark cycle (lights on 8 a.m. to 8 p.m.) and provided with standard rodent chow diet (Prolab RMH 1800, LabDiet) and water ad libitum. The animals were monitored for general health score (1–5; NZ Animal Health Care Standard), weight, and food intake three times per week. When the study was completed (day 16) the animals were killed by carbon dioxide inhalation overdose followed by cervical dislocation.

### 4.3. Experimental Design

The study design consisted of control and loperamide-treated groups, with 8 males and 8 females per group (control males *n* = 8, control females *n* = 8, loperamide-treated males *n* = 8, loperamide-treated females *n* = 8), thus 32 rats in total. The experiment was conducted in four blocks of 8 rats consisting of 4 rats from each treatment group (control or loperamide; 2 males and 2 females per treatment group in a block). The female loperamide treatment group had *n* = 7 rather than *n* = 8 because one animal had to be culled immediately prior to the study due to non-treatment-related health reasons.

Following acclimatisation, the animals were handled 5 min every day for one week prior to behavioural tests by an experimenter to minimise stress during the experiment (Figure 1a). A two-dose protocol was followed to ensure that the loperamide effect (4 h half-life in rats) did not subside due to metabolism by the liver over 48 h between testing and sampling [62]. A dose was selected that demonstrably inhibits colonic motility via the ENS in rats [63], yet does not overwhelm the P-glycoprotein efflux system to cross the blood–brain barrier [64]. On day 15, rats were administered subcutaneously with DMSO (control) or loperamide (4 mg/kg) in DMSO (10%) two hours prior to the first behaviour test which was the OF test (Figure 1b). On day 16 rats were re-administered with control or loperamide two hours prior to sampling (Figure 1c).

### 4.4. Behaviour Tests

On the day of behaviour testing, animals were acclimatised for 30 min prior in the test room. The OF and EPM test protocols and video analysis followed methods previously described [65]. The OF test was used to assess anxiety-like exploratory and locomotor behaviours over 10 min. The EPM (Panlab, Harvard Apparatus, Holliston, MA, USA) was used to test anxiolytic/anxiogenic effects of the drug over 5 min. The number of rearings was also measured through manual detection. Recorded videos were analysed using EthoVision XT 10 (Noldus, Wageningen, The Netherlands) and the experimenters were blinded to treatment.

### 4.5. Sample Collection

Brain (hippocampus, prefrontal cortex, and the amygdala), proximal colon tissue, and caecal content were collected immediately postmortem. Samples were stored in RNAlater (Ambion, Life Technologies, Carlsbad, CA, USA) for 24 h and then transferred to −20 °C for gene expression analysis. Caecal content for microbiome analyses was snap frozen in liquid nitrogen and stored at −80 °C.

### 4.6. Gene Expression

#### 4.6.1. RNA Isolation

Isolation of RNA from brain and proximal colon tissue was performed using QIAzol Lysis Reagent (Qiagen, Valencia, CA, USA) followed by clean up using a RNeasy Lipid Tissue Mini Kit (Cat. No. QLAG74804, Qiagen, Valencia, CA, USA), according to the manufacturer’s instructions.

#### 4.6.2. RNA Analysis Using RNA-Seq

Strand-specific cDNA libraries were prepared using NEBNext^®^ Ultra Directional RNA Library Prep Kits for Illumina^®^ (New England Biolab, Ipswich, MA, USA). The libraries were size-selected to include fragments of 250–300 bp and were sequenced using an Illumina Novaseq 6000, generating 150 bp paired-end sequences. Trimmomatic 0.36 was employed for quality trimming of the reads, and the remaining read pairs that passed the quality trimming were aligned to the Rattus norvegicus genome (Rnor 6.0 release 102) using STAR 2.7.4a [66]. Uniquely mapped read pairs were then aggregated for each gene and subjected to analysis using a likelihood ratio generalised linear model in the EdgeR 3.42.0 package for R [67]. Genes exhibiting a >1.5-fold difference (i.e., |log fold change| > 0.58) with a false discovery rate (FDR) < 0.05 were considered differentially expressed. Additionally, gene expression was assessed through gene set enrichment analysis (GSEA) using the mroast function from limma [68], with Reactome pathways serving as the gene sets. In GSEA, the collective expression of groups of genes is examined as a single unit rather than analysing individual genes independently [69].

### 4.7. Caecal Microbiota

DNA extraction and metagenomic sequencing were performed as previously described [70]. Briefly, metagenomic DNA was extracted using Nucleospin Soil kits (Macherey-Nagel GmbH, Duren, Germany) with bead beating, libraries were prepared using the NEBNext Ultra DNA Library Prep Kit, and sequencing was performed using an Illumina Novoseq 6000 [61]. Sequence reads were quality trimmed using Trimmomatic, read pairs merged using PEAR, 0.9.6 and host sequences removed using the bbudk.sh function from the BBMAP 38.80 package [61].

The “blastx” function of DIAMOND version 0.9.22 was used to map reads against the “nr” NCBI database to assign taxonomy [71]. Permutation multivariate analysis of variance of the microbiome composition was performed using the ANOSIM function from the Vegan 2.6-4 package for R [72]. To assign putative functions to the DIAMOND alignment files against the KEGG database [73], MEGAN6 Ultimate Edition was utilised. Differences in the relative abundances of individual taxa and gene functions were analysed using the MaAsLin2 1.13.0 package in R [74].

### 4.8. Statistical Analysis

Animal metric and behavioural data were analysed using GraphPad Prism (version 9.5.1, GraphPad software Inc., San Diego, CA, USA). Metric data were analysed using repeated measures two-way ANOVA (sphericity not assumed; Greenhouse–Geisser correction applied) followed by Sidak’s test for multiple comparisons post hoc.

Behavioural data was analysed using two-way ANOVA (sphericity not assumed; Greenhouse–Geisser correction applied) followed by Tukey’s multiple comparison test post hoc (*p* < 0.05). Data are presented as mean ± standard error of mean (SEM).

Integrated analysis was performed on the behaviour, gene expression, and microbiota datasets. Sparse projection to latent structures models with sparse discriminant analysis (sPLS-DA) were used to find correlated variables [75].

## 5. Conclusions

Our findings implicate an anxiogenic gut to brain signaling effect via ENS modulation associated with stress-related CRH and GABA gene pathways in the hippocampus in females and thus demonstrate the importance of routinely incorporating both males and females in research. This has clinical significance for understanding the relationship between gut motility and anxiety-related mood conditions such as irritable bowel syndrome and investigation of new pharmacological ENS targets that alter gut to brain signalling to modulate mood. It may also prompt consideration of possible mood effects following use of Imodium™ and other opioids.

## Figures and Tables

**Figure 1 ijms-25-13183-f001:**
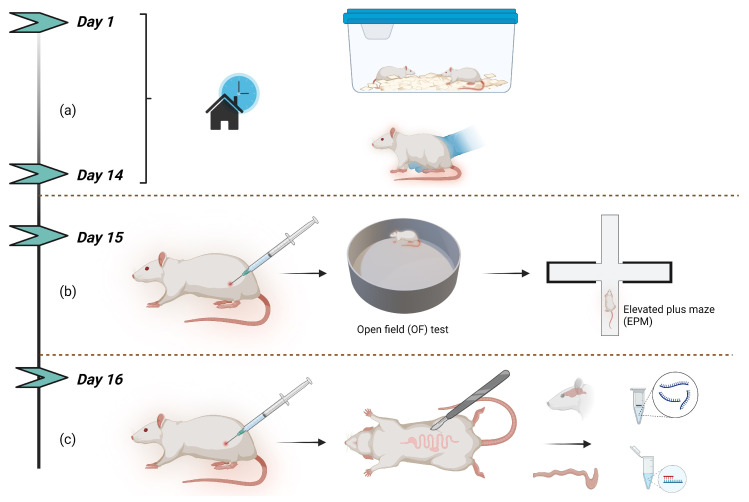
Study design: (**a**) Rats were acclimatised to their new living environment for one week after which they were handled for one week; (**b**) On the day of the behaviour tests, rats were administered with loperamide or DMSO (control) two hours prior to the start of the behaviour testing (OF, EPM); (**c**) Rats were re-administered with loperamide or DMSO (control) the next day, two hours prior to sampling. Created in BioRender. (2024) BioRender.com/x26y637.

**Figure 2 ijms-25-13183-f002:**
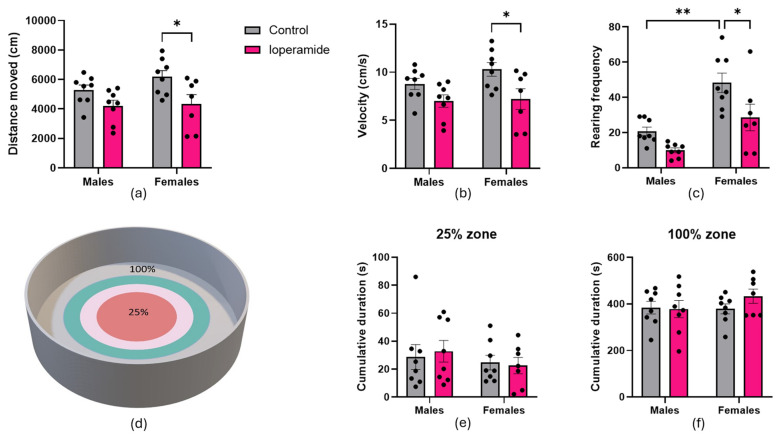
Open field test: (**a**) Distance moved; (**b**) Velocity of tracked movement; (**c**) Rearing frequency; (**d**) Coloured concentric circles are representative of different zones in the arena (red represents center or 25% zone; grey represents periphery or 100% zone); (**e**) Time spent in 25% or center zone of OF arena; (**f**) Time spent in the 100% zone or periphery of the OF arena. Asterisks indicate statistical significance (* *p* < 0.05; ** *p* < 0.01). Data shown as mean with error bars indicating SEM, *n* = 7–8 animals per treatment group.

**Figure 3 ijms-25-13183-f003:**
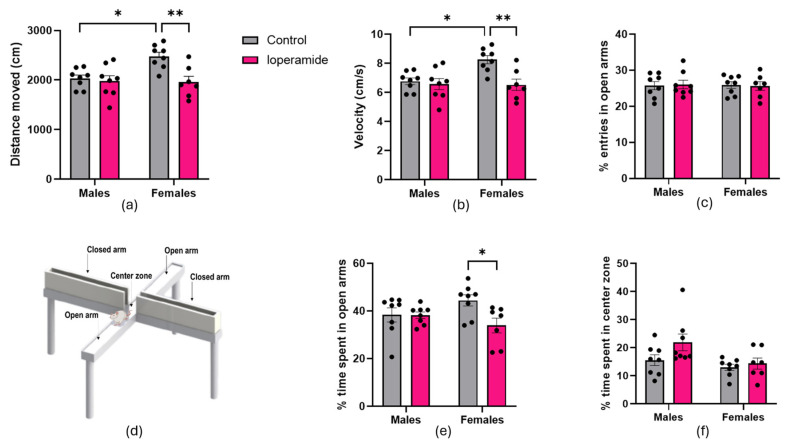
Elevated plus maze: (**a**) Distance moved; (**b**) Velocity of tracked movement; (**c**) Percent entries in open arms of the EPM; (**d**) Diagram of the EPM; (**e**) Graph showing % time spent in open arms of the EPM; (**f**) Percent time spent in center zone of the EPM. Asterisks indicate statistical significance (* *p* < 0.05; ** *p* < 0.01). Data shown as mean with error bars indicating SEM, *n* = 7–8 animals per treatment group.

**Figure 4 ijms-25-13183-f004:**
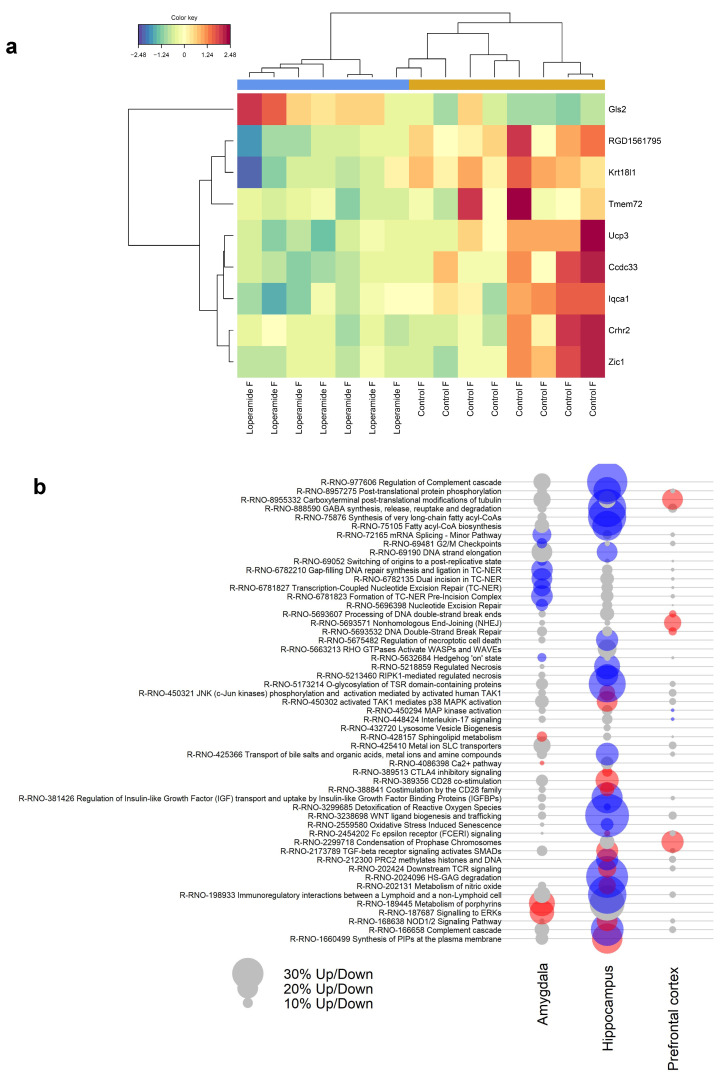
(**a**) Heatmap showing differentially expressed genes in the hippocampus of loperamide-treated (*n* = 7) and control (*n* = 8) female rats. The red and blue colour scale represents expression, with red being higher and blue being lower. The values are scaled by row, which means the actual expression (counts) has been converted to standard deviations above and below the median which is set at zero; (**b**) Reactome pathways differentially expressed by gene set enrichment analysis (*p* < 0.05) in amygdala, hippocampus, and prefrontal cortex of female rats. Red circles indicate overall significantly higher expression in loperamide rats compared to controls and blue circles indicate overall significantly lower expression compared to controls. Grey circles indicate pathways not differentially expressed (*p* > 0.05). The size of circles is proportional to the number of up- or downregulated genes.

**Figure 5 ijms-25-13183-f005:**
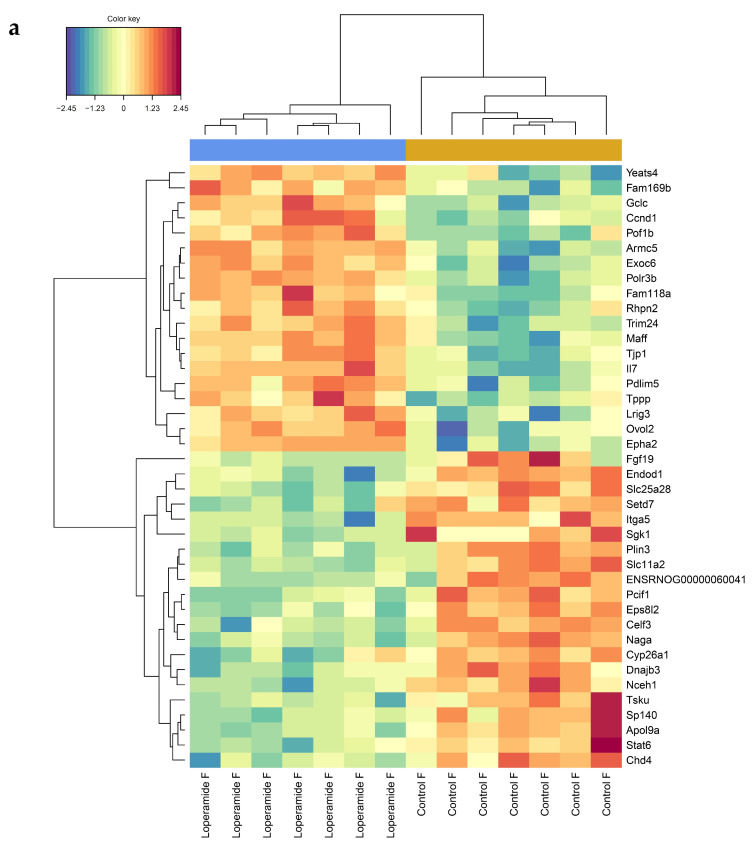
(**a**) Volcano plot of control versus loperamide-treated groups for differentially expressed genes in the proximal colon. Heatmaps showing the top 40 differentially expressed genes in the proximal colon of (**b**) female (*n* = 7) and (**c**) male rats (*n* = 8). The red and blue colour scale represents expression, with red being higher and blue being lower. The values are scaled by row, which means the actual expression (counts) has been converted to standard deviations above and below the median which is set at zero.

**Figure 6 ijms-25-13183-f006:**
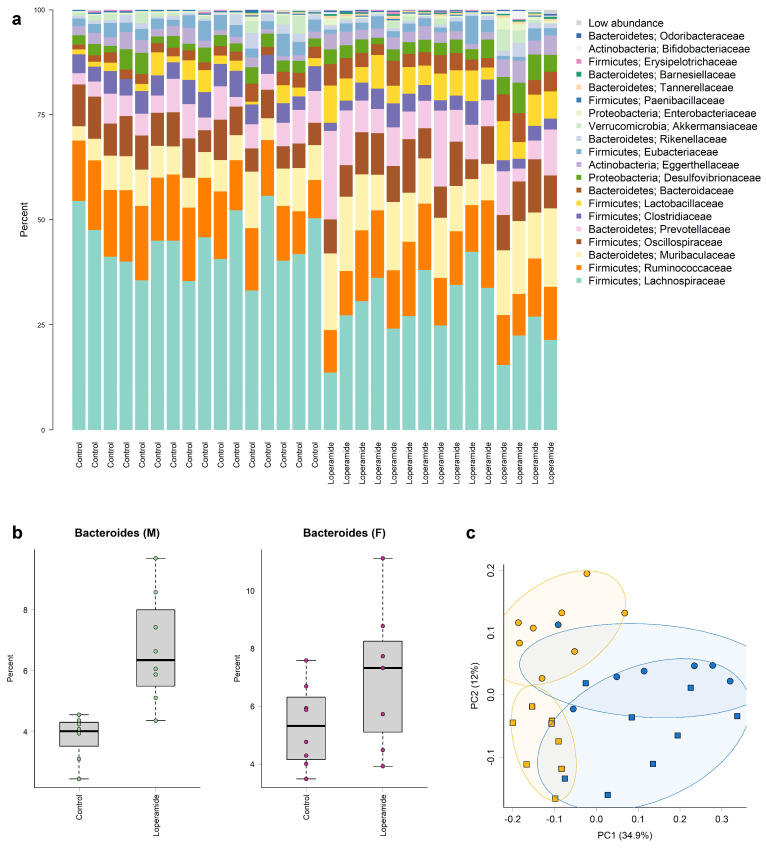
(**a**) Taxonomic composition of the caecal microbiota at the family level. The *x* axis represents treatment, and the *y* axis represents relative abundance in percent. Low-abundance groups are the sum of all taxa outside of the 20 most abundant families. (**b**) Principal coordinate analysis (PCoA) plot of weighted UniFrac phylogenetic distances of caecal microbiotas from control (yellow) or loperamide (blue) groups, *n* = 8 males (squares) and 8 females (circles) per treatment group (PC1 vs. PC2). Percentages on axes indicate the proportion of variation explained by each dimension. Permutation analysis of variance indicated a significant difference between loperamide and control communities (ANOSIM *p* value = 0.001, R statistic = 0.449), ellipse depicts 75% confidence interval. (**c**) Box plots showing Bacteroides to be more abundant in loperamide-treated male and female rats compared to controls (median with 95% confidence intervals).

**Figure 7 ijms-25-13183-f007:**
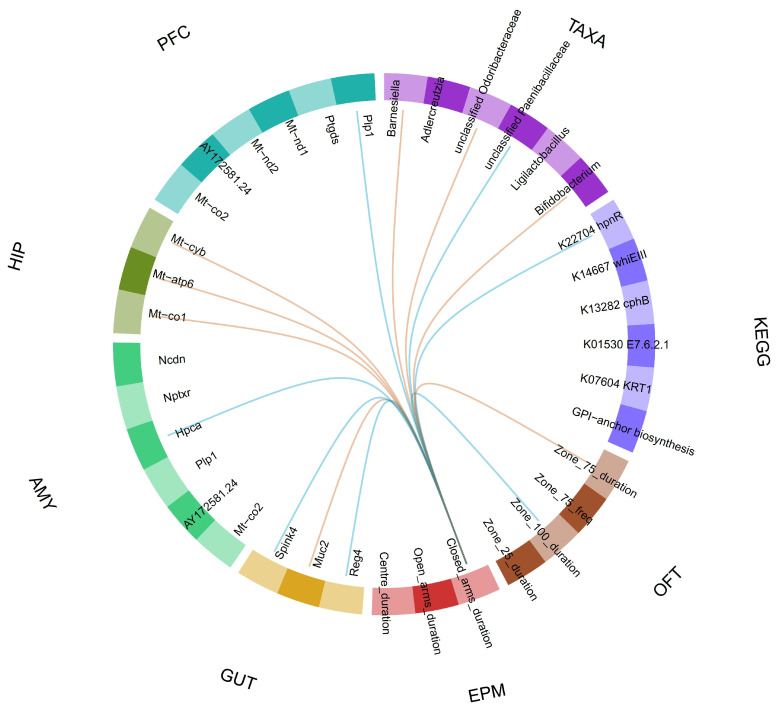
Correlation between time spent in the closed arm of the EPM and other variables identified by the sPLS-DA algorithm. Variables are displayed in the circle grouped by the type of variables for: genes from PFC; prefrontal cortex; HIP, hippocampus; AMY, amygdala; GUT, proximal colon; TAXA are the different caecal microbiome taxa; KEGG are microbiome genes/KEGG orthologues; EPM, parameters from the elevated plus maze; OFT, parameters from the open field test. Variables with a correlation score > 0.75 are joined by an orange line and variables with a correlation score < −0.75 are joined by a blue line.

**Table 1 ijms-25-13183-t001:** Effect of loperamide on rat body weight, food intake, and faecal output.

Sex	Treatment	Bodyweight (g)		Food Intake (g)		Faecal Output (g)		Rat
		Pre	Post	Pre	Post	Pre	Post	*n*
Male	Control	481.62 ± 14.30	480.75 ± 13.82	32.12 ± 1.79	31.37 ± 1.08	9.61 ± 0.46	9.18 ± 0.61	8
	Lop.	480.87 ± 16.20	474.87 ± 17.54	33.37 ± 1.20	20.75 ± 2.01 ***	9.62 ± 0.44	6.68 ± 0.53 ***	8
Female	Control	266.87 ± 8.64	265.12 ± 8.22	21.87 ± 1.07	19.12 ± 0.76	5.93 ± 0.24	5.71 ± 0.22	8
	Lop.	278.85 ± 8.44	275.42 ± 8.65	22.85 ± 0.67	12.85 ± 1.71 **	6.08 ± 0.16	3.80 ± 0.45 **	7

Posttreatment (day 16) was compared with pretreatment (day 15) in the same animal. Mean values for each treatment group ± SEM, *n* = 31 rats in total. ** *p* < 0.01; *** *p* < 0.001. Lop.: loperamide.

**Table 2 ijms-25-13183-t002:** Differentially expressed genes in the hippocampus of female rats.

Gene	Protein	logFC	FDR	*p* Value	Lop.
*Tmem72*	Transmembrane protein 72	−5.63141	0.04781366	2.61 × 10^−5^	Decreased
*Crhr2*	Corticotropin releasing hormone receptor 2	−4.0482	0.04781366	4.18 × 10^−5^	Decreased
*RGD1561795*	Unknown function	−2.71571	0.04781366	3.90 × 10^−5^	Decreased
*LOC683212*	Similar to keratin complex 1, acidic gene 18	−2.39166	0.04781366	1.79 × 10^−5^	Decreased
*Ucp3*	Uncoupling protein 3	−2.29239	0.04781366	3.17 × 10^−5^	Decreased
*Zic1*	Zic family member 1	−2.19697	0.04781366	2.27 × 10^−5^	Decreased
*Ccdc33*	Coiled-coil domain containing 33	−1.58709	0.04781366	1.99 × 10^−5^	Decreased

Based on significance (FDR < 0.05 and |logFC| > 0.585). Lop.: loperamide.

**Table 3 ijms-25-13183-t003:** Taxa with significantly different relative abundance between control and loperamide-treated male and female rats.

Phylum	Family	Genus	Male		Female		*q*-Value	
			Control	Lop.	Control	Lop.	Trtmt	Sex
Bacteroidetes	Bacteroidaceae	Bacteroides	3.83 ± 0.25	6.72 ± 0.62	5.33 ± 0.50	7.01 ± 0.95	0.004	0.180
Bacteroidetes	Barnesiellaceae	Barnesiella	0	0.42 ± 0.12	0.14 ± 0.09	0.44 ± 0.12	0.004	0.415
Bacteroidetes	Muribaculaceae	Duncaniella	0	0.58 ± 0.09	0.33 ± 0.13	0.49 ± 0.13	0.009	0.303
Bacteroidetes	Muribaculaceae	Muribaculum	0.77 ± 0.12	1.31 ± 0.11	1.19 ± 0.13	1.35 ± 0.16	0.041	0.143
Bacteroidetes	Tannerellaceae	Parabacteroides	0	0.73 ± 0.13	0.21 ± 0.14	0.71 ± 0.13	<0.001	0.416
Firmicutes	Lactobacillaceae	Lactobacillus	3.66 ± 1.01	9.93 ± 0.91	4.38 ± 1.09	8.78 ± 1.03	0.001	0.971
Firmicutes	Lactobacillaceae	Limosilactobacillus	0.21 ± 0.14	0.44 ± 0.14	0.32 ± 0.16	1.11 ± 0.11	0.016	0.076
Firmicutes	Clostridiaceae	Butyricicoccus	2.16 ± 0.13	1.54 ± 0.20	2.34 ± 0.24	1.76 ± 0.25	0.016	0.429
Firmicutes	Clostridiaceae	Clostridium	5.82 ± 0.38	4.15 ± 0.68	7.05 ± 0.68	3.77 ± 0.72	0.003	0.767
Firmicutes	Incertae sedis	Colidextribacter	2.51 ± 0.06	1.74 ± 0.16	1.80 ± 0.14	1.65 ± 0.23	0.026	0.066
Firmicutes	Eubacteriaceae	Eubacterium	4.47 ± 0.39	2.98 ± 0.43	6.56 ± 0.81	4.55 ± 1.44	0.026	0.176
Firmicutes	Lachnospiraceae	Acetatifactor	2.82 ± 0.23	1.35 ± 0.19	2.34 ± 0.23	1.19 ± 0.20	<0.001	0.339
Firmicutes	Lachnospiraceae	Blautia	1.61 ± 0.05	1.05 ± 0.12	1.74 ± 0.13	1.02 ± 0.16	<0.001	0.930
Firmicutes	Lachnospiraceae	Dorea	1.61 ± 0.16	1.06 ± 0.14	1.95 ± 0.15	1.12 ± 0.11	<0.001	0.278
Firmicutes	Lachnospiraceae	Enterocloster	1.38 ± 0.08	0.92 ± 0.09	1.42 ± 0.09	0.90 ± 0.11	<0.001	0.951
Firmicutes	Lachnospiraceae	Roseburia	5.22 ± 0.63	3.36 ± 0.54	4.56 ± 0.51	2.71 ± 0.61	0.011	0.347
Firmicutes	Lachnospiraceae	Schaedlerella	2.59 ± 0.30	1.51 ± 0.19	3.69 ± 0.40	1.97 ± 0.28	<0.001	0.066
Firmicutes	Ruminococcaceae	Flavonifractor	1.32 ± 0.02	0.94 ± 0.08	0.98 ± 0.08	0.92 ± 0.13	0.041	0.111
Firmicutes	Ruminococcaceae	Ruminococcus	6.14 ± 0.20	5.04 ± 0.49	5.39 ± 0.27	4.17 ± 0.28	0.008	0.105
Proteobacteria	Enterobacteriaceae	Klebsiella	0.67 ± 0.03	0.15 ± 0.10	0.40 ± 0.12	0.23 ± 0.11	0.008	0.411

Data represented as mean percent ± standard error of mean (SEM), *n* = 8 males and 7–8 females per treatment group. Lop.: loperamide, Trtmt; treatment.

**Table 4 ijms-25-13183-t004:** Relative abundance (percentage) of top 5 caecal microbiome function pathways.

**Male**					
**Level 1**	**Level 2**	**Level 3**	**Control**	**Lop.**	***q*-Value**
Metabolism	Nucleotide Metabolism	K1000240 Pyrimidine metabolism	2.615 ± 0.015	2.849 ± 0.028	<0.001
Metabolism	Nucleotide Metabolism	K1000230 Purine metabolism	6.904 ± 0.054	7.293 ± 0.054	0.010
Metabolism	Carbohydrate Metabolism	K1000010 Glycolysis/Gluconeogenesis	2.780 ± 0.020	2.950 ± 0.025	0.010
Environmental Information Processing	Signal Transduction	K1002020 Two- component system	6.910 ± 0.136	5.664 ± 0.195	0.010
Metabolism	Amino Acid Metabolism	K1000300 Lysine biosynthesis	1.346 ± 0.014	1.433 ± 0.011	0.011
**Female**					
**Level 1**	**Level 2**	**Level 3**	**Control**	**Lop.**	***q*-Value**
Metabolism	Xenobiotics Biodegradation and Metabolism	K1000361 Chlorocyclohexane and chlorobenzene degradation	0.037 ± 0.001	0.051 ± 0.001	0.015
Metabolism	Nucleotide Metabolism	K1000230 Purine metabolism	6.758 ± 0.070	7.219 ± 0.090	0.049
Metabolism	Carbohydrate Metabolism	K1000040 Pentose and glucuronate interconversions	1.598 ± 0.028	1.429 ± 0.031	0.049
Metabolism	Metabolism of Other Amino Acids	K1000480 Glutathione metabolism	0.051 ± 0.004	0.076 ± 0.004	0.049
Metabolism	Carbohydrate Metabolism	K1000052 Galactose metabolism	3.208 ± 0.069	2.838 ± 0.060	0.049

Based on significance; *q* < 0.05. KEGG database functional gene annotation at classification level 3. Lop.: loperamide. *n* = 7–8 animals per treatment group.

## Data Availability

The original data presented in the study are openly available upon reasonable request from AgResearch at PRJNA1127265.

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
