# Peer review of "Peripherally Restricted Activation of Opioid Receptors Influences Anxiety-Related Behaviour and Alters Brain Gene Expression in a Sex-Specific Manner"

_ijms, 2024, doi:10.3390/ijms252313183_

Round 1
Reviewer 1 Report
Comments and Suggestions for Authors
This is an interesting study about the role of peripheral opioid receptors in gut-brain signaling. The assessment of sex-dependence is advantageous.
There are however several concerns that need to be addressed.
1. The title, experimental approach, data, and discussion/conclusions do not account for the possible involvement of delta- and/or kappa-opioid receptors in the loperamide-mediated effects.
2. The authors attributed the suppressed locomotor activity to heightened anxiety. However, this may not be the case. It is very possible that the anxiety-like behavior is the result of decreased locomotor activity. Loperamide resulted in reduced in locomotor activity in both the OFT and the EPM which suggests induction of sedation and/or sickness-like behavior. Overall, the data do not conclusively indicate a pure anxiolytic effect of loperamide.
3. The Results section should provide more statistical data related to the 2-way ANOVA (i.e., F, df, p for main effects and interactions).
4. The gene expression/correlation analyses are relatively complex and many readers are likely not familiar with interpretation of the data/figures (particularly Fig. 7). Therefore, additional clarification and interpretation of the figures would be beneficial.
5. How does data in Fig. 7 fit with data in Fig. 2 (which indicates no differences in duration in the 100% zone?
6. Fig. 7 legend states correlation with "distance moved" in EPM, yet text in section 2.6 regarding Fig. 7 describes "duration in closed arm"
7. The report would benefit from further analysis of key findings obtained in the various multivariable analyses.
Author Response
Thank you for taking the time to review our manuscript. We hope our responses to the questions raised provide the necessary information and clarity.
Reviewer#1:
This is an interesting study about the role of peripheral opioid receptors in gut-brain signaling. The assessment of sex-dependence is advantageous.
There are however several concerns that need to be addressed.
Question 1. The title, experimental approach, data, and discussion/conclusions do not account for the possible involvement of delta- and/or kappa-opioid receptors in the loperamide-mediated effects.
Response 1. Loperamide demonstrates a significantly higher affinity and selectivity for μ-opioid receptors compared to δ- and κ-opioid receptors, with Ki (inhibition constant) values of 3, 48, and 1156 nM, respectively. Since a lower Ki value indicates stronger binding affinity, this suggests that μ-opioid receptors are the primary target for loperamide's effects. While the potential involvement of δ- and κ-opioid receptors is an interesting avenue for future research, it was beyond the scope of this study to investigate their contribution.
Question 2. The authors attributed the suppressed locomotor activity to heightened anxiety. However, this may not be the case. It is very possible that the anxiety-like behavior is the result of decreased locomotor activity. Loperamide resulted in reduced in locomotor activity in both the OFT and the EPM which suggests induction of sedation and/or sickness-like behavior. Overall, the data do not conclusively indicate a pure anxiolytic effect of loperamide.
Response 2. Loperamide is known to have limited central nervous system penetration due to its inability to cross the blood-brain barrier, suggesting that its effects on behavior are primarily peripheral. The observed reduction in locomotor activity was specific to female rats and was not observed in males. If loperamide were inducing sedation or central effects, we would expect similar reductions in locomotor activity in both sexes, which was not the case here.
In addition to the behavioral differences, our molecular data further support the idea that the observed changes may not be due to sedation. Our Gene Set Enrichment Analysis (GSEA) revealed a significant downregulation of the GABA synthesis, release, reuptake, and degradation pathway in the hippocampus of female rats treated with loperamide. Since sedative drugs typically enhance GABAergic activity, the downregulation of this pathway further suggests that the reduced locomotor activity is not due to sedation. Rather, it may reflect peripheral effects that could contribute to the observed anxiety-like behavior.
Moreover, we observed altered brain gene expression, particularly of CRHR2, a gene associated with the stress response, which was specifically altered in female rats. This supports these changes as contributing to the behavioural differences observed.
Question 3. The Results section should provide more statistical data related to the 2-way ANOVA (i.e., F, df, p for main effects and interactions).
Response 3. This has been added to Figure 2 (line 110-111) and Figure 3 (line 131-133) legends.
Question 4. The gene expression/correlation analyses are relatively complex and many readers are likely not familiar with interpretation of the data/figures (particularly Fig. 7). Therefore, additional clarification and interpretation of the figures would be beneficial.
Response 4. Figure legend updated to read:
Figure 7. Correlation between time spent in the closed arm of the EPM and other variables identified by the sPLS-DA algorithm. Variables are displayed in the circle grouped by the type of variables for: genes from PFC; prefrontal cortex, HIP; hippocampus, AMY; amygdala, GUT; proximal colon, TAXA are the different caecal microbiome taxa, KEGG are microbiome genes/KEGG orthologs, EPM; parameters from the elevated plus maze, OFT; parameters from the open field test. Variables with a correlation score > 0.75 are joined by an orange line and variables with a correlation score < -0.75 are joined by a blue line.
Question 5. How does data in Fig. 7 fit with data in Fig. 2 (which indicates no differences in duration in the 100% zone?
Response 5. Fig 7 relates to how different variables correlate to each other, irrespective of groups. Fig 2 shows how rats in different groups (M vs F, treatment vs control) performed in the OFT.
Question 6. Fig. 7 legend states correlation with "distance moved" in EPM, yet text in section 2.6 regarding Fig. 7 describes "duration in closed arm".
Response 6. The legend for Figure 7 has now been changed to reflect Question 4 above and now refers to ‘time spent in the closed arm of the EPM’.
Question 7. The report would benefit from further analysis of key findings obtained in the various multivariable analyses.
We have added further discussion on mitochondrial genes associated with EPM correlation in relation to anxiety, with 3 new refs. (Lines 470-473).
Lower brain mitochondria respiration rate has been reported in highly anxious rats and is consistent with our findings [54,55] and is consistent with our findings that time spent in the closed arms positively correlated with expression of three hippocampal mitochondrial genes (Mt-cyb, Mt-atp6, Mt-co1) involved in energy metabolism. Human studies show increasing evidence for a link between mitochondrial function and stress-related anxiety [56,57].
Reviewer 2 Report
Comments and Suggestions for Authors
1. Line 75-76: I am curious why caecum microbiome was chosen for sequencing, but not the whole area corresponding to the proximal colon tissue? There is quite a bit variation between caecum microbiome and colon microbiome.
2. Figure 4B: Why conducting GSEA on prefrontal cortex and amygdala since there were no differences at the individual transcript level. Why there will be numbers of genes up or down regulated in prefrontal cortex and amygdala indicated by size of circle, if there was no difference at gene level to begin with? Additionally, for GSEA analysis, (FDR) < 0.05 should be used instead of p <0.05 for individual pathways.
3. Please include a full list of genes from EdgeR analysis for brain and colon as supplemental tables (Excel sheet), not just DEGs.
4. 2.4: Adding volcano plots can help visualize large amount of DEGs.
5. Line 188-192: no figures were shown related to this part of discussion.
6. Table 3: is the q value for treatment refers to both male and female or is there separate q value for female and male? And please include the coefficient from Masalin output.
7. 2.5.1: One of the advantages of shotgun metagenomic sequencing over 16S sequencing is that you can get resolution to the species level. I think the authors should focus on the species and even strain level in the microbiome analysis.
8. Line 239-248: Where is figure 6d and 6e mentioned in the text?
9. Figure 6b is missing legend. And what statistics were used? The figure legend says Permutation analysis of variance but in the parentheses says ANOSIM. Regardless of the method, the author should also provide statistics for pairwise comparison.
10. Figure 6c: Statistics is missing in both figure and the main text.
11. 2.5.2: The authors did not conduct pathway analysis. What authors used might be KEGG Orthology (KO) identifier, not Kegg pathway id. Additionally, KO id usually start with the letter “K” followed by a five-digit number, so I do not even know what it is listed here.
12. 4.6.2 please provide citation the software/package that were used. Also, what package was used to count genes after STAR?
13. 4.7 Please describe how taxonomy was assigned for bacteria species. How beta diversity was analysis is not described. Similarly, cite all software/package.
14. Data availability: I am not able to locate the data using PRJNA1127265 at AgResearch. Please provide more details.
Author Response
We thank the reviewer for taking the time to review our manuscript and hope our responses to the questions raised provide the necessary information and clarity.
Reviewer#2:
Question 1. Line 75-76: I am curious why caecum microbiome was chosen for sequencing, but not the whole area corresponding to the proximal colon tissue? There is quite a bit variation between caecum microbiome and colon microbiome.
Response1.
The caecum was chosen because it is guaranteed to have content for analysis whereas the proximal region/area of the colon may not always have content for analysis and we did not want to risk missing data. Because the proximal colon is very short in the rat we expect the least variability in content composition between caecum and this colon region.
Question 2. Figure 4B: Why conducting GSEA on prefrontal cortex and amygdala since there were no differences at the individual transcript level. Why there will be numbers of genes up or down regulated in prefrontal cortex and amygdala indicated by size of circle, if there was no difference at gene level to begin with? Additionally, for GSEA analysis, (FDR) < 0.05 should be used instead of p <0.05 for individual pathways.
Response 2. The GSEA is used precisely when significant differences between genes at the individual are not observed. The theory behind GSEA is that the collective shift in a group of related genes (e.g., those within a certain pathway) can be more informative that a shift in any one gene. The GSEA method accounts for this. Similarly, P values rather than FDR is appropriate here as we are highlighting an overall shift in transcriptome, rather than looking at any one gene.
https://academic.oup.com/nar/article/43/7/e47/2414268
https://www.pnas.org/doi/10.1073/pnas.0506580102
Question 3. Please include a full list of genes from EdgeR analysis for brain and colon as supplemental tables (Excel sheet), not just DEGs.
Response 3. While we can provide this, the data is currently across many tables/files and would take some time to get it into a presentable Excel format. We therefore prefer to make this available on request.
Question 4. 2.4: Adding volcano plots can help visualize large amount of DEGs.
Response 4. We have added a volcano plot for the proximal colon (Figure 5a).
Question 5. Line 188-192: no figures were shown related to this part of discussion.
Response 5. We have referred here (line 188) to the added volcano plot for the proximal colon (Figure 5a).
Question 6. Table 3: is the q value for treatment refers to both male and female or is there separate q value for female and male? And please include the coefficient from Masalin output.
Response 6. These are separate Q values for sex and treatment as shown in Table 3. We do not have the coefficient values available at this time, as we would need to rerun our analyses.
Question 7. 2.5.1: One of the advantages of shotgun metagenomic sequencing over 16S sequencing is that you can get resolution to the species level. I think the authors should focus on the species and even strain level in the microbiome analysis.
Response 7. Species level is still not overly reliable for shotgun sequencing. Roughly only half the taxa can be assigned to species level, which is why we have aggregated at genus level.
Question 8. Line 239-248: Where is figure 6d and 6e mentioned in the text?
Response 8. These are errors and have been corrected to 6a-c in the text.
Question 9. Figure 6b is missing legend. And what statistics were used? The figure legend says Permutation analysis of variance but in the parentheses says ANOSIM. Regardless of the method, the author should also provide statistics for pairwise comparison.
Response 9. ANOSIM is a type of permutation multivariate analysis of variance. In the statistics section, we have added (line 566-568 ref# 69) “Permutation multivariate analysis of variance of the microbiome composition was performed using the ANOSIM function from the Vegan package for R.” And the following reference: https://www.jstor.org/stable/3236992
Question 10. Figure 6c: Statistics is missing in both figure and the main text.
Response 10. See Table 3, row 1. This is now referred to in the text (line 249). We have also added to the Figure 6c legend “(median with 95% confidence intervals)” for clarity.
Question 11. 2.5.2: The authors did not conduct pathway analysis. What authors used might be KEGG Orthology (KO) identifier, not Kegg pathway id. Additionally, KO id usually start with the letter “K” followed by a five-digit number, so I do not even know what it is listed here.
Response 11. While the reviewer is technically correct in that we are looking at KEGG Orthologies, it is reasonable to refer to them as pathways, as per wording from KEGG themselves “The KO (KEGG Orthology) database is a database of molecular functions represented in terms of functional orthologs. A functional ortholog is manually defined in the context of KEGG molecular networks, namely, KEGG pathway maps, BRITE hierarchies and KEGG modules, and is given a KO identifier called K number”.
KO ids were provided in the text and table (Table 4 column 3).
Question 12. 4.6.2 please provide citation the software/package that were used. Also, what package was used to count genes after STAR?
Response 12. These references have been added to section 4.6.
Reference for STAR (Dobin et al., 2013) (line 549, ref #63) https://pmc.ncbi.nlm.nih.gov/articles/PMC3530905/
Reference for EdgeR (Robinson et al., 2009) (line 551, ref #64) https://pmc.ncbi.nlm.nih.gov/articles/PMC2796818/
Reference for GSEA (mroast function in limma) (Ritchie et al., 2015) (line 554, ref #65) https://academic.oup.com/nar/article/43/7/e47/2414268
STAR itself generates the gene/transcript counts
Question 13. 4.7 Please describe how taxonomy was assigned for bacteria species. How beta diversity was analysis is not described. Similarly, cite all software/package.
Response 13. Information and reference have been added. (Line 565-566, ref #68)
The “blastx” function of DIAMOND version 0.9.22 was used to map reads against the “nr” NCBI database to assign taxonomy.
Reference: Buchfink B, Xie C, Huson DH. Fast and sensitive protein alignment using diamond. Nat Methods. (2015) 12:59–60. doi: 10.1038/nmeth.3176
Question 14. Data availability: I am not able to locate the data using PRJNA1127265 at AgResearch. Please provide more details.
Response14. Accession number is for NCBI, not AgResearch. Data will become publicly accessible upon publishing.

Round 2
Reviewer 1 Report
Comments and Suggestions for Authors
The authors addressed to some extent all 7 concerns and the revised manuscript is improved. However, questions 1-3 still remain.
1. The relative affinity and selectivity of loperaminde for the various opioid receptors certainly point to mu as a key site of action. However, these characteristics do not provide conclusive evidence that the observed effects are in fact due to solely to actions at MOR. Thus, the title and conclusions remain overstated.
2. The authors suggest that "if loperamide were inducing sedation and central effects, we would expect similar reductions in locomotor activity in both sexes," What is the basis/evidence for this expectation? It still cannot be ruled out that the observed anxiety-like behavior is secondary to decreased locomotor activity. Also, the OFT time in 100% zone data showing no differences among groups further argues against pronounced anxiety-like behavior.
3. The Results section still does not provide information on main effects and interactions of main effects.
Author Response
Reviewer#1:
Thank you for the opportunity to provide further information in our responses. We apologise for not being sufficiently explicit in our initial response. We have revised our responses and incorporated these into the manuscript.
This is an interesting study about the role of peripheral opioid receptors in gut-brain signaling. The assessment of sex-dependence is advantageous.
There are however several concerns that need to be addressed.
Question 1. The title, experimental approach, data, and discussion/conclusions do not account for the possible involvement of delta- and/or kappa-opioid receptors in the loperamide-mediated effects.
Response 1. Loperamide demonstrates a significantly higher affinity and selectivity for μ-opioid receptors compared to δ- and κ-opioid receptors, with Ki (inhibition constant) values of 3, 48, and 1156 nM, respectively. Since a lower Ki value indicates stronger binding affinity, this suggests that μ-opioid receptors are the primary target for loperamide's effects. While the potential involvement of δ- and κ-opioid receptors is an interesting avenue for future research, it was beyond the scope of this study to investigate their contribution.
Response 1 extra.
We have removed “mu” opioid receptor from the title.
We now describe loperamide as “a potent μ-opioid receptor agonist…” in the abstract.
We have added this to our introduction (line 69): Loperamide (Imodium) is an antimotility drug that activates ENS opioid receptors primarily of the mu type, and also delta and kappa but with lower affinity [24], in the myenteric plexus of the large intestine, decreasing activity of the myenteric plexus leading to a reduction in tone of the longitudinal and circular smooth muscles and a slowing of GI transit [25,26].
We have added this to our discussion (line 372): “Loperamide primarily activates mu opioid agonist in the ENS but will also have acted on delta and kappa receptors located on enteric neurons throughout the rat GI tract, and these will likely also have contributed to the opioid receptor mediated effects.”
https://pmc.ncbi.nlm.nih.gov/articles/PMC6310692/ J. J. Galligan and C. Sternini 2017
Question 2. The authors attributed the suppressed locomotor activity to heightened anxiety. However, this may not be the case. It is very possible that the anxiety-like behavior is the result of decreased locomotor activity. Loperamide resulted in reduced locomotor activity in both the OFT and the EPM which suggests induction of sedation and/or sickness-like behavior. Overall, the data do not conclusively indicate a pure anxiolytic effect of loperamide.
Response 2. Loperamide is known to have limited central nervous system penetration due to its inability to cross the blood-brain barrier, suggesting that its effects on behavior are primarily peripheral. The observed reduction in locomotor activity was specific to female rats and was not observed in males. If loperamide were inducing sedation or central effects, we would expect similar reductions in locomotor activity in both sexes, which was not the case here.
In addition to the behavioral differences, our molecular data further support the idea that the observed changes may not be due to sedation. Our Gene Set Enrichment Analysis (GSEA) revealed a significant downregulation of the GABA synthesis, release, reuptake, and degradation pathway in the hippocampus of female rats treated with loperamide. Since sedative drugs typically enhance GABAergic activity, the downregulation of this pathway further suggests that the reduced locomotor activity is not due to sedation. Rather, it may reflect peripheral effects that could contribute to the observed anxiety-like behavior.
Moreover, we observed altered brain gene expression, particularly of CRHR2, a gene associated with the stress response, which was specifically altered in female rats. This supports these changes as contributing to the behavioural differences observed.
Response 2 extra.
We have added at line 408: “We cannot rule out the possibility that the increase in anxiety is due to the decrease in locomotor activity. However, it seems unlikely that loperamide at this dose would have acted centrally to cause sedation and have affected only females. While morphine can induce sedation, this occurs for both sexes and the effect is greater in males than in females.”
https://pubmed.ncbi.nlm.nih.gov/17217999/ (ref 35)
Question 3. The Results section should provide more statistical data related to the 2-way ANOVA (i.e., F, df, p for main effects and interactions).
Response 3. This has been added to results sections.
Response 3 extra.
2.1 (line 86)
Statistical analysis using 2-way repeated measures ANOVA showed main effects of Loperamide treatment (F(1,27) = 33.29, p<0.0001) and sex (F(1,27) = 47.39, p<0.0001) for food intake, and Loperamide treatment (F(1,27) = 21.22, p<0.0001) and sex (F(1,27) = 44.00, p<0.0001) for fecal output. There was no interaction between sex and treatment for any of the animal metrics.
2.2.1 (line 111)
Statistical analysis using 2-way repeated measures ANOVA showed a main effect of
Loperamide treatment for distance moved (F(1,27) = 10.29, p = 0.0034) and for velocity, (F(1,27) = 10.29, p = 0.0034)). Main effects of treatment (F(1,27) = 10.73, p = 0.0029) and sex (F(1,27) = 24.86, p<0.0001)) and were found for rearing frequency. No interaction between treatment and sex were found for any of the animal metrics.
2.2.2 (line 138)
Statistical analysis using 2-way repeated measures ANOVA showed main effects of Loperamide treatment (F(1,27) = 8.804, p = 0.0062) and sex (F(1,27) = 4.900, p = 0.0355) for distance moved in the EPM, in addition to an interaction between Loperamide treatment and sex (F(1,27) = 5.686, p = 0.0244). Main effects of Loperamide treatment (F(1,27) = 8.811, p = 0.0062) and sex (F(1,27) = 4.890, p = 0.0355) were found for velocity in the EPM, in addition to an interaction between Loperamide treatment and sex (F(1,27) = 5.686, p = 0.0244). A main effect of Loperamide treatment (F(1,27) = 4.536, p = 0.0425) but not sex (F(1,27) = 0.1262, p = 0.7252) was found for time spent in the open arms, in addition to an interaction between Loperamide treatment and sex (F(1,27) = 4.265, p = 0.0486).
Reviewer 2 Report
Comments and Suggestions for Authors
Author Response
(No reply requested, but box seems to need to be filled in to allow revised manuscript upload.)
Round 3
Reviewer 1 Report
Comments and Suggestions for Authors
much improved